# Strategic Imperatives of Managing the Sustainable Innovative Development of the Market of Educational Services in the Higher Education System

**Inna Gryshova [1], Nataliia Demchuk [2], Iryna Koshkalda [3,\*] , Nataliia Stebliuk [4] and Nataliia Volosova [5]**

1   JSNU-SPbPU Institute of Engineering-Sino-Russian Institute, Jiangsu Normal University, 101 Shanghai Rd, Tongshan Qu, Xuzhou 221100, Jiangsu, China; 6020180146@jsnu.edu.cn
2   Department of Finance, Banking and Insurance, Dnipro State Agrarian and Economic University, 49600 Dnipro, Ukraine; natademchyk@gmail.com
3   Department of Land Administration and Cadastre, Kharkiv National Agrarian University named after V.V. Dokuchayev, 62483 Kharkiv, Ukraine
4   Department of Management of Organizations and Administration Dniprovsk State Technical University, 51900 Kamianske, Ukraine; tasha-s@ukr.net
5   Department of Higher Mathematics Dniprovsk State Technical University, 51900 Kamianske, Ukraine; volosonata@ukr.net
\*   Correspondence: irinavit1506@gmail.com

**Abstract:** The conditions of the functioning of state higher education institutions at the present stage of development of the Ukrainian economy require new approaches to solve the problem of the relationship between the volume of training of specialists with higher education and their employment in the sphere of economic activity. The purpose of this article was to provide theoretical substantiation and practical recommendations for the development of a higher education institution development strategy for making managerial innovative decisions on balancing the demand and supply of educational services in a competitive environment. The following methods were used in the study: abstract logical; comparative economic and system structural; statistical; sociological; modeling; the algorithm of practical application of the theory of constraints of systems; and the apparatus of the theory of fuzzy sets. Methodological approaches to the implementation of the optimal allocation of budget places of the university by means of a practical application of system restriction theory and fuzzy set theory were proposed. The result was the allocation of budgetary places, taking into account the demand for specialties in terms of the economic situation of the region, the demand in the labor market, the demand among entrants, and the proposal of the institution of higher education and the Ministry of Education and Science of Ukraine. It will make it possible to reach the optimum balance between demand for specialists in specific specialties and their supply. The practical value of the research results lies in the development and use of methodological provisions for forecasting the demand and supply of educational services of higher education institutions, which are means of prospective reflection, predicting the ways of further development of the higher education system and modeling of various options for its functioning.

**Keywords:** development strategy; economic-mathematical model; marketing of educational services; innovative decisions; formal logic; theory of fuzzy sets; theory of system limitations

## 1. Introduction

The balance of the needs of the educational services market and the labor market is a positive sign of the stability and efficiency of the functioning of the higher education system in the region and the country as a whole. Therefore, at the present stage of development of the educational environment, it is necessary to use scientific tools to understand market patterns of interaction of supply and demand for educational services, which will not only make it possible to navigate consumer preferences, but also to shape them. In such circumstances it is vital to investigate changes in the demand for educational services in order to adapt them to the higher education system.

Scientists propose to determine the prospective need of the state for specialists with a certain level of qualification to introduce a limit of the total number of places for admission of students to higher education, based on staffing and financial information and other resources, with the distribution of licensed volumes of admission among the best institutions on a competitive basis. So, it is advisable at this stage to set no more than half of the number of people who complete upper secondary education each year (this approach is used in Sweden). For most countries in the West, "state procurement" is formed, first of all, on the basis of real needs of the economy, and the needs of the individual are met at the expense of the individual. In Germany, they do not train specialists in any specialty, unless there is a firm belief in the need for such specialists in the labor market. In the United Kingdom, quotas are set for the training of ministries—health, defense, and education.

In Ukraine, the definitions of higher education institutions, ministries, and agencies on the state procurement of specialist training are conditional and indicate the absence of a well-established mechanism of interaction between higher education institutions, employment centers, and statistical authorities, which leads to an overestimation of the region's professional needs.

Innovative development of education should first of all be implemented through the prism of such educational and qualification training of specialists, which would meet the potential needs of regional labor markets with a focus on the knowledge-intensive types of economic activity.

In this regard, the methodological and practical issues of balancing the demand and supply of education services in higher education are becoming important and relevant. The problems of identifying issues and specifics of regional educational markets remain unresolved in scientific works. This necessitates research into the market for educational services and forecasting the needs of higher education professionals, both in the short and long term. This will make it possible to reach the optimum balance between demand for specialists in specific specialties and their supply.

The purpose of this article is to provide theoretical substantiation and practical recommendations for the development of a higher education institution development strategy for making managerial innovative decisions on balancing the demand and supply of educational services in a competitive environment.

To achieve this goal, the following tasks were set:

— to analyze the main tendencies of development of the educational services and labor market in Ukraine;
— to build an economic and mathematical model using statistical analysis, an algorithm for practical application of system constraint theory (TOC), a "fuzzy" set theory apparatus, which will allow the determination of the optimal number of students of a certain specialty;
— provide guidance on balancing supply and demand for educational services to make strategic management decisions.

The following methods were used in the study: abstract logical; comparative economic and system structural; statistical; sociological; modeling; the algorithm of practical application of the theory of constraints of systems; and the apparatus of the theory of fuzzy sets.

This article is structured as follows. The introduction explains the research background and determines the main goal of the study. In Section 2 contains the literature review of the topic and an interpretation of the results obtained.

Section 3 focuses on the materials and methods used with regard to the research methodology, the chosen sampling, data collection, and analysis methods. The empirical results of the study are presented in Section 4. In Section 5, the main findings are summarized, and the limitations and future research are outlined.

## 2. Literature Review

A review of the literature and a search for articles related to the current study were conducted. The time period (2004–2019) was chosen based on the large amount of literature published during this time. A total of 39 articles were found through a systematic review of the literature and analyzed for the complexity and interdependence of managerial problems in higher education that require new ideas and approaches, which necessitates the search for new management solutions.

The research on the market of educational services and activities of higher education institutions (universities) was devoted to the work of such scientists as Kratt [1], who examined the market of higher education services in terms of the following points of view: the first point reflects the time parameter of the activity of the higher education system; the second point corresponds to the product sector parameter of higher education activity; the third one corresponds to the territorial parameter of the structure of the operational field of economic research. Krasovskaya [2] proposed a marketing approach to assessing the international competitiveness of national higher education systems, which is based on the analysis of quantitative and qualitative characteristics of the market. Lelyk [3] examined the impact on education services of key economic indicators related to the provision of educational services in higher education institutions and proposed to plan the volume of educational services at the state level and to regulate the training of specialists for the national economy. Basha [4] proposed comprehensive marketing research on the market of educational services of national universities, which was based on the segmentation of target groups of consumers and buyers of educational services, with the identification of graduates' needs for the commercialization of their diploma and dissertation research results.

Researchers believe that the provision of innovative educational development should first and foremost be carried out through the prism of such educational qualification training that would meet the potential needs of regional labor markets with a focus on knowledge-intensive economic activities [5–9].

Research has been conducted on the expectations and requirements of employers for competencies of graduates of institutions of higher economic education [10,11]. The obtained results became the basis for determining the prospects for innovations in the marketing of economic educational services, orienting graduates at certain behavioral models in the process of study [12]. Labanauskis, et al. [13], examined the relationships and importance of the components (criteria) composing the study process at the university level. The article aimed to reveal the diversity of the study process and evaluate the importance and significance of the criteria composing it. The rest of the criteria do not mean less importance of the quality of the study process at the university level [14]. Despite that, paying attention to a certain scenario of criteria regarding the strategy and allocation of resources can lead to unique institutional performance and achievements in the quality of the study process at the university level. The companies' representatives, as well as students, recommend that practical subjects be incorporated in the course of study. The results support the assumption that the current generation of university students studies to succeed in the future, but it has relatively high expectations about work and career that may reduce its employability if it does not have the appropriate work experience and social habits [15–18].

The well-functioning links between universities and companies and knowledge transfer in supporting the economic growth are becoming increasingly more important. Successful partnership between employers and universities will ensure relatively sustainable development of higher economic education, on the one hand, and, on the other hand, improve the image and reputation of enterprises, which means their sustainable development and competitiveness. This will confirm the importance and

demand of the enterprise as a mandatory partner for improving the quality of higher education [19,20]. Analysis of employers, taking into account their readiness to form partnership relationships with HEIs, will contribute to improving the quality of training personnel in business structures and HEIs. The modern world should focus on the formation of graduates' competencies in accordance with the demands of employers, which will provide greater access to practical skills for students throughout the entire period of university studies [21].

Currently, higher education is facing global forces that require innovative research, innovative pedagogy, and innovative organizational structures. The formation of a new, relevant, modern, world-class national innovation system of education cannot be solved without understanding the role of education and identifying key areas of transformation that have been proven based on the actual state and development prospects [22]. Polozhentseva's [23] article contains an analysis of universities' global rankings based on the selection of indicators of KPI-monitoring system to show the development of university competitive strengths related to countries' economic development. Semenets-Orlova et al. [24] conducted an analysis of scientific approaches to understanding organizational development in the field of educational institution management as a complex of successive educational changes.

The main factors determining the quantitative changes of higher schools' students' contingent have been stated. The data envelopment analysis method was applied, considering the number of enrolled students, budget financing, co-financing, and self-financing as inputs, and the number of graduated students according to the field of education as the output [25,26].

Despite the importance and value of systematic studies conducted in the domestic and foreign economies, new approaches to the formation of the state procurement mechanism and defining the competitive strategy of higher education organizations require further development. The effectiveness of the strategy implementation of higher education development depends, to a large extent, on the choice of strategic alternatives, the coordination of strategic priorities and goals, which necessitates the improvement of the methodological principles of strategic management (Tarasenko I. et al., 2018) [27] (Lysytsia, N. et al., 2017) [28]. These processes are caused by the need to adapt the system of higher education and professional training to dynamic changes in today's globalized world.

The conducted research gives grounds to assert that it is expedient to solve the urgent problem of choosing the optimal strategies of higher education organizations with the help of modern approaches and methods of economic and mathematical modeling.

When constructing economic and mathematical models to formalize descriptions of fuzzy systems and to process research results, one of the most effective approaches is to apply the system constraint theory and fuzzy set theory. In the face of intuitive fuzzy parameters and certain limitations, Yu and Zhang [29,30] demonstrated the effectiveness and applicability of the proposed approach.

## 3. Material and Method

In today's conditions of the development of society, knowledge and information become the main resources and source of wealth as an individual, and the state as a whole. The competitiveness of the country's economy is determined not by the volume of natural or productive resources, but primarily by intellectual potential, the ability to generate new knowledge. The level of development of the market of educational services in the country is one of the determining indicators of its competitiveness. The market for educational services in Ukraine is developing in the context of the general laws of the economy market, but at the same time, it has a number of specific features. In particular, its high dynamism, territorial segmentation, and local character, high rate of capital turnover, high sensitivity of educational services to market conditions due to the impossibility of their storage and transportation, and individuality of production [31]. The market relationships of the production of educational services have specific characteristics related to state interference and regulation of the most significant services, as well as restrictions on private entrepreneurial activity. Education as a service calls for fulfilling student needs for acquiring knowledge through learning specific and specialized programmes that fit industry requirements and satisfy the consumer. The consumer or students' needs are the primary

focus, hence all marketing activities should be directed at them. However, education services are intangible with no ownership rights and should be instrumental in fulfilling a gap in industry or country. There is the need to strategically plan the marketing and management of tertiary institutions, taking cognizance of the nature and characteristics of services and the implication for the education sector. Also marketing plans should be directed at building reputation and boosting student inflows [32]. Educational institutions today serve as "producers" and "sellers" of educational services, and pupils, parents, and students are consumers, and act as actors in the market of educational services. Also, directly or indirectly, HEIs are in contact with other actors—enterprises and organizations employing graduates of higher educational institutions, financial foundations, publishing houses of magazines, personnel agencies, and the country (Figure 1).

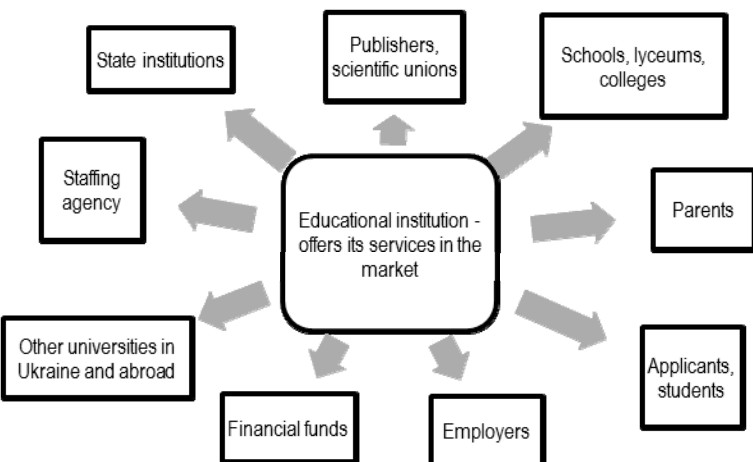

**Figure 1.** Subjects of the market of educational services. Sources: authors' development based on [33].

Each of these subjects implements certain economic interests: students are interested in obtaining a balance in terms of investment of time, energy, and effort in learning and the result at the end of the training process as specialists; prospective employers are interested in graduates being able to solve problems commonly faced by the organization that they will work in; the society has its own expectations regarding the proper functioning of the economy also in the social plan due to the effort of the graduates of higher education both during the initial training and during continuous training; the state—to create such conditions in the market of educational services and the education sector as a whole, to ensure an increase in the share of highly skilled specialists in the economic system, and to improve the quality of human capital [34]. Therefore, entities form a proposal for educational services, while others demand.

Another component of the state of the market for educational services is the offer. Educational institutions offer education services, which offer educational services for students, and for graduates and the labor market [35]. New elements of society, especially business, develop faster, while the other parts of society react slowly to any changes. In particular, infrastructure is incapable of supporting the university–enterprise partnership, which is demanded by the knowledge-based market economy. Achieving the matching of supply and labor supply on the labor market will reduce one of the major socio-economic problems on the path to economic growth—unemployment. From this perspective, it is important to coordinate market activities among themselves: the education system should direct its activities to the needs of the economy, in particular, to the needs of the labor market. The optimal functioning of the investigated markets is achieved through harmonization of the interests of the subjects of these markets.

The subjects of the labor market and educational services are the country, educational institutions, employers, and workers, the relationships between which are depicted in Figure 2.

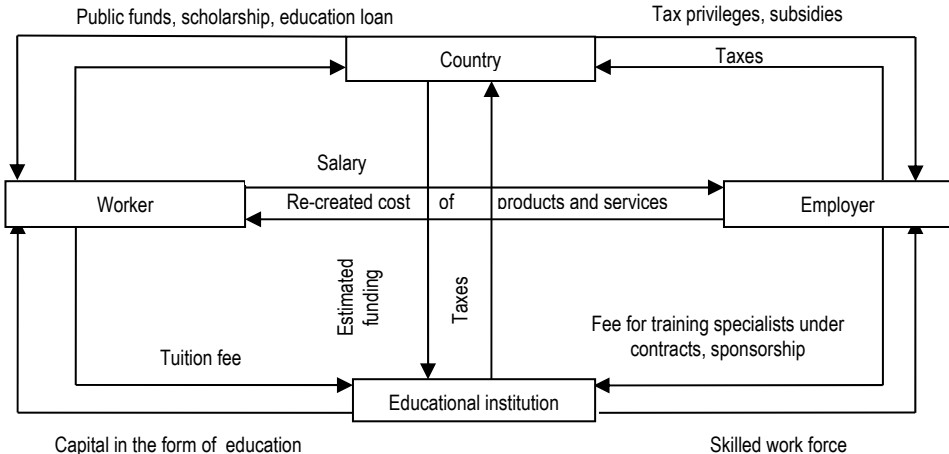

**Figure 2.** The relationship of the main actors of labor markets in educational services. Sources: authors' development based on [35].

The advantage of the relationship between the subject of the labor market and educational services lies in the deep integration of educational resources. However, the roles and motivation of the parties involved are different. Only through upholding the win–win approach and meeting their expectations can the cooperative training mode develop sustainably.

On the one hand, all involved parties should make their own position clear, and not be offside or absent in the cooperation. The government and education departments should be committed to creating a favorable environment for cooperation, emphasizing policy design for cooperation, rather than interfering with the operation and implementation of cooperative education. Enterprises are expected to offer postgraduates timely information on market demands and enable them to select research topics closely linked to industrial application. Universities are supposed to act as the main body in innovative postgraduate training, while the research institutes which are not qualified for postgraduate training should play a collaborative role and share the technical advantage through project cooperation. Employers, as the platform for testing the quality of scientific achievements and talent training, are required to give timely feedback for the sake of training scheme optimization. On the other hand, all parties should cooperate closely to build an open channel for information exchange and sharing. In so doing, they can integrate their own resources and give full play to their respective advantages [8,36].

During the last nine years, the total number of public procurement sites has undergone tangible changes, but their distribution between industries has remained virtually constant. The largest volumes of government procurement fall into such industries as social sciences, economics, law, humanities, and engineering, production, and construction, and the smallest for pedagogical sciences, services, medicine, and agriculture. In addition, 51% of them study at their own expense (that is, funds of individuals); these funds are disposed of directly by educational institutions (retrieved from http://www.ukrstat.gov.ua) [37]. Among the priority areas, according to the contractual form of study, were the same economics, law, and humanities, but physical education and art also emerged. The smallest number of entrants desires to pay for such areas as aviation, rocket-space and marine engineering, electronics, metallurgy, and materials science.

Confirmation of the imbalance of the state and the entrant is shown in the priorities for the specialties, as shown in Figure 3.

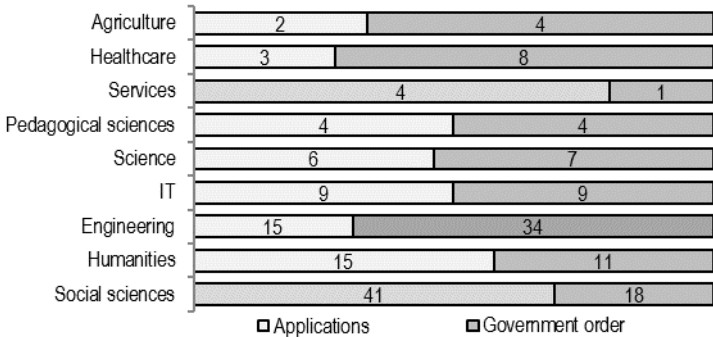

**Figure 3.** The correspondence of the number of baccalaureate applications and the state order for bachelor education, full-time, 2015. Sources: authors' development based on [38].

As can be seen from the figures of Figure 3, there is a significant imbalance between such branches as social sciences and services, engineering and agriculture. In the ratio of entrants, the first group choose prestigious specialties, despite the fact that the state does not need a large number of specialists. In contrast, the specialties that are less prestigious for entrants are more of a state priority.

Higher educational establishments mainly focus on the demand of consumers of educational services, which leads, on the one hand, to the growth of quantitative and qualitative indicators of training specialists and an overestimation of the labor market by specialists of "fashion" professions, and, on the other, to an increase in unemployment among graduates of educational institutions.

Hence, a higher school in Ukraine does not fulfill its main goal: it constantly lags behind the demands of the labor market, preparing specialists who were needed several years ago. The negative image of the system of vocational education is affected by the unemployment rate of the graduates, and the poor qualifications of graduates from employers.

Such negative tendencies are caused by the subjective interests of higher educational institutions interested in the quantitative indicators of the contingent of students with which the level of financing of higher educational institutions is connected, both from the state budget and extra budgetary funds. Although the labor market is looking for representatives of today's unprincipled technical professions of the middle and lower levels and technical specialties in general, the proper teaching profession is poorly promoted and remains insufficiently attractive to young people, since not only does it require more long-term and complex training, but it also does not meet contemporary ideas about prestige of work and comfortable living conditions [39].

In addition, the employer is rather passive and in fact does not take part in the order of specialists and improve the quality of educational services, namely, their interest primarily affects the training of specialists, because it depends on them for further employment. Although employers themselves, in almost all organizations and private structures, prefer experienced people who have acquired certain skills rather than graduates of higher education institutions with only theoretical knowledge.

This study targeted the Dnipropetrovsk region as one of the most economically developed regions in Ukraine. It is characterized by favorable geographical location, rich natural resources, powerful industrial and scientific potential, developed agricultural production, and a high level of transport and communications development.

Prospects (directions of development) of the Dnipropetrovsk region:

- Expansion, reconstruction, technical and technological re-equipment of existing production in order to increase the competitiveness of products and strengthen the competitive position in the domestic and foreign markets, introduction of modern energy-saving, environmentally friendly technologies;
- Implementation of international and European standards (obtaining appropriate certificates of quality management systems) at food and processing industry enterprises, expansion of their range;

- Conducting the development of agricultural service cooperatives—the creation and development of cooperatives in different fields of activity:　milk, grain, fruit and vegetable, meat processing, multifunctional;
- Implementation of the established municipal council with the general plan for a promising development of the regional center, based on which new modern quality requirements for living creatures and the future development of the site are formed;
- Tourism development (green, industrial tourism, ethnographic routes) and sports (development of sports and recreational complexes and sports grounds).

In Table 1 presents the demand of specialists in the labor market of Dnipropetrovsk region according to the website rabota.ua as of 18 October 2017.

**Table 1.** Rating of vacancies in Dnipropetrovsk region.

| Position | Number of Vacancies | Position | Number of Vacancies |
|---|---|---|---|
| Sales Manager | 1178 | Waiter | 73 |
| Customer Service Manager | 1178 | Maid | 70 |
| Seller | 383 | Lawyer | 63 |
| Seller-consultant | 383 | Regional manager | 59 |
| Driver | 304 | Chief Accountant | 54 |
| Sales Representative | 302 | Courier | 53 |
| Administrator | 214 | Secretary | 40 |
| Cashier | 214 | PC operator | 38 |
| Loader | 165 | Medical representative | 37 |
| Accountant | 160 | Economist | 32 |
| Storekeeper | 115 | System administrator | 32 |
| Office Manager | 111 | Head assistant | 27 |
| Guardian | 97 | Bartender | 25 |
| Director | 84 | Merchandiser | 18 |

Sources: authors' development.

The main problems of the educational services market are:

- An inefficient, overly centralized, outdated system of management and financing of the internal sphere of education;
- An effective system of distribution of public procurement satisfies the needs of students who are interested only in free study without regard for the university or specialty;
- A non-transparent process of planning and distribution of volumes of state orders among universities, the privilege of certain higher educational institutions receiving an increased level of financing according to special norms makes the system of public procurement difficult to predict for universities themselves;
- A mismatch of vacancies and qualifications and absence of an effective system of employment for specialists;
- A lack of a well-established mechanism of interaction between education and enterprises, trade and culture;
- The dependence of educational services on the applicants' demands;
- Uncertainty about the qualifications of applicants, the absence of a system for determining their inclination and abilities.

Consequently, the maintenance of the multi-year status quo in conditions of dynamic changes in the labor market leads to a deepening of differences between universities and the economy. This once again confirms the thesis about the need to move away from the model of government order and look for other forms of distribution of budget funds.

In Table 2 shows the vacancies for specialists with basic higher education (the rating was determined by the number of vacancies for this specialty), according to the Municipal Employment Center (as of 18 October 2017), in Kamianske.

**Table 2.** Rating of vacancies in Kamianske.

| Position | Number of Vacancies | Position | Number of Vacancies |
|---|---|---|---|
| Customer Service Manager | 56 | Guardian | 4 |
| Sales Manager | 56 | Regional manager | 3 |
| The driver | 16 | Maid | 3 |
| Seller-consultant | 15 | Chief Accountant | 3 |
| Seller | 15 | Secretary | 2 |
| Sales Representative | 14 | Waiter | 2 |
| Loader | 11 | PC operator | 2 |
| Cashier | 10 | Courier | 2 |
| Storekeeper | 7 | System administrator | 1 |
| Accountant | 7 | Medical representative | 1 |
| Administrator | 7 | Economist | 1 |
| Office Manager | 5 | Director | 1 |
| Lawyer | 4 | | |

Sources: authors' development.

For collecting primary data, a survey method was used for which a questionnaire was developed. For the survey, a sample of 156 students of all faculties of Dniprovsk State Technical University DSTU was formed, who were studying in various specialties and were graduates of city schools. The survey was conducted by means of personal interviews and online questionnaires in social networks.

For generalization and processing of the results of the questionnaire, a program for working with Excel spreadsheets was used.

In conditions of high instability of socio-economic processes, higher education institutions are forced to overcome a lot of contradictions. One of the most effective approaches to escape from their limits is the theory of system constraints (theory of constraints, TOC), developed in the 1970s by the Israeli physicist Dr. Eliah M. Goldtratt [40]. The construction of the management decision-making process on the basis of the TOC involves the concentration of organizational resources to eliminate constraints (conflicts) that prevent the institution from fully realizing its potential.

The algorithm for the practical application of TOC consists of four main steps, which are based on a systematized and focused approach [40]:

1.  Definition of the limitation of the system;
2.  Solution (how to use this restriction);
3.  Subordination of all processes to the decision;
4.  Removing the identified restriction, expanding its capabilities.

If in the previous stages, the "restriction" is eliminated (ceased to be a "constraint"), return to the initial stage (step 1) of study system. The results of the production system will be determined by other constraints. Work on optimization needs to be replicated under the new conditions [41]. Define the main goals for implementing the strategy:

$S_1$—financial resources;

$S_2$—material-technical base, information resources;

$S_3$—educational–pedagogical and scientific potential;

$S_4$—optimal allocation of budget places in specialties.

Thus, we had four qualitative parameters that described the initial state of the external environment ($S_0$) and the desired state ($S_t$). The vector of the strategic solution s = ($s_1$, $s_2$, $s_3$, $s_4$) determined the value for the key indicators of the implementation of the strategy.

However, there was the problem that, along with indicators that were characterized by quantitative values, it was necessary to complete the description of the state and analysis of qualitative indicators. When processing data using the mechanisms of formal logic there is a contradiction between fuzzy knowledge and clear methods of logical output. You can solve this problem by using special methods of presentation and processing of fuzzy data [41]. An apparatus of fuzzy set theory was applied, which had the following features: it could formalize the dependencies of almost any complexity, the parameters in fuzzy models may be different, to describe the dependencies between parameters use natural language, fuzzy models have high adaptability to expert data. To evaluate each parameter, the apparatus of the theory of fuzzy sets was used. This made it possible to evaluate the specified parameters within the interval (0; 1) of real numbers, and then perform their linguistic recognition. The mathematical apparatus of fuzzy set theory and fuzzy logic greatly facilitates the formalization of the problems of preference logic and allows you to successfully solve complex decision-making problems in the context of uncertainty, uncertainty and subjectivity of estimates.

The main results of the research were the development of methodological and mathematical support for the implementation of the optimal allocation of budget places of higher education institutions by the proposed method, including the means of the theory of system constraints and the theory of fuzzy set. The concept of demand dynamics modeling for the services of higher education institutions using modern approaches and methods of economic and mathematical modeling is offered, which, unlike the existing ones, gives the opportunity to make decisions about the application of managerial influence to achieve the desired level of demand.

## 4. Results and Discussions

The processing and analysis of the questionnaires yielded the following results. It was discovered that most students choose their specialty on the advice of parents and university staff, but one fifth of the students, even enrolled in the university, did not identify with their future profession. This indicates that not all graduates can adequately assess what they have the ability to do and what they want. In addition, more than half of the respondents were not exactly sure that they had made the right choice. In order to assist in choosing a specialty, it would be advisable to introduce a methodology for competently evaluated graduates. The need for the practice of implementing this methodology was confirmed by the results of the survey, which showed that 63% would use a professional consultant on professional issues. Due to a lack of awareness of their skills and poor awareness of the situation in the labor market, more than half of those surveyed were interested in humanities, which at the time were not in demand. Most importantly, almost 70% of entrants considered prestige the most important criterion when choosing a school of education.

Therefore, the problem of choosing a profession and an institution of higher education is one of the most important problems for graduates, both in Kamianske and in Ukraine as a whole. After all, the choice of profession for a young person is the basis of their self-affirmation in society: who to be, what social group to belong to, and what goals to achieve. As a result of the analysis of the questionnaire it should be noted that not every senior student is ready to make a choice consciously. It was found that parents, relatives, and friends had the greatest influence on the choice of higher education institutions and specialties. However, in addition to the immediate environment, the choice of a senior pupil was influenced by the prestige of a university, its specialty. Unfortunately, few people choose a profession based on the demands of the labor market, the possibility of real employment. Higher education advertising and university employees have a fairly small percentage of influence. As a result, most students are not completely satisfied with their chosen specialty. A high percentage of respondents, 30%, reported that if they chose universities again, they would have enrolled in another. Among specialties, those which most interested students were humanitarian, and only later were they interested in technical ones. The reason for this tendency is the very low level of prestige of technical professions in the country.

In the first step of the algorithm for the practical application of TOS implementation, the system constraint was defined—$s_4$-distribution of budget places in specialties. The main problem established in step 2 was the inconsistency between the demand for a specialty in terms of the economic situation in the region, demand in the labor market, demand among entrants, and the offer of universities and MES. For analysis and the possibilities of further numerical processing, weighted weight coefficients $k_{ij}$ were determined for the main branches of the economy (Table 3).

**Table 3.** Value of weight coefficients for industries.

| № | Economic Sector | Weight Direction Ratio | | | |
|---|---|---|---|---|---|
| | Industry | Condition in the Area | Labor Market | University | Applicant |
| 1 | Extractive Processing: | 0.5 | 0.4 | 0.4 | 0.2 |
| | • Metallurgical production, production of finished metal products | 0.4 | 0.4 | 0.3 | 0.2 |
| | • Production of food products | 0.4 | 0.3 | 0.3 | 0.2 |
| | • Production of chemicals and chemical products | 0.3 | 0.2 | 0.3 | 0.1 |
| | | 0.2 | 0.2 | 0.3 | 0.1 |
| | • Production of rubber and plastic products; other non-metallic mineral products | 0.6 | 0.5 | 0.4 | 0.2 |
| | Engineering | 0.2 | 0.2 | 0.3 | 0.1 |
| | Supply of electricity | | | | |
| 2 | Agriculture | 0.5 | 0.3 | 0.3 | 0.1 |
| 3 | Construction | 0.4 | 0.3 | 0.4 | 0.2 |
| 4 | Trade | 0.3 | 0.3 | 0.4 | 0.4 |
| 5 | Entrepreneurship | 0.5 | 0.5 | 0.4 | 0.5 |
| 6 | Transport and communications | 0.4 | 0.5 | 0.5 | 0.4 |
| | Education | 0.6 | 0.6 | 0.5 | 0.3 |
| 7 | Healthcare | 0.4 | 0.4 | 0.3 | 0.3 |
| | Culture | 0.3 | 0.1 | 0.2 | 0.1 |
| | Sport | 0.2 | 0.2 | 0.1 | 0.1 |
| 8 | Tourist and resort | 0.2 | 0.2 | 0.1 | 0.3 |
| 9 | Foreign | 0.4 | 0.3 | 0.2 | 0.4 |

Sources: authors' development.

Graphically, for estimating the situation for each sector of the economy, the values of the weight coefficients obtained can be represented as a pyramid (Figure 4).

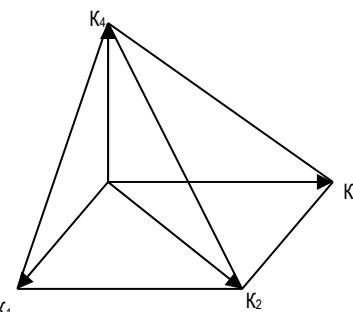

**Figure 4.** Pyramid of distribution of values of weight coefficients by a certain branch. Sources: authors' development.

At the heart of the pyramid is a quadrilateral, built on vectors whose lengths are the coefficients in terms of the economic condition of the region, demand in the labor market, and the offer of universities and MES, and at the top, the value of the demand factor of the specialty among entrants as the main factor from the point of view of the university. In step 3, the optimal allocation of budget places was

determined. To calculate the integral index for each direction by the nature of the output data, it is expedient to use the formula of the mean geometric value (1):

$$k_i = \sqrt[n]{\prod_{j=1}^{n} k_{ij}}. \tag{1}$$

Table 4 shows the calculated values of integral indicators by sectors of the economy.

**Table 4.** Calculation of the values of the integral indicator by branches of economy.

| № | Economic Sector | Integral Indicator |
|---|---|---|
| 1 | **Industry** | |
| | Extractive | $K_1 = 0.355656$ |
| | Processing: | |
| | • Metallurgical production, production of finished metal products | $K_2 = 0.31301$ |
| | • Production of food products | $K_3 = 0.291295$ |
| | • Production of chemicals and chemical products | $K_4 = 0.205977$ |
| | • Production of rubber and plastic products; other non-metallic mineral products | $K_5 = 0.186121$ |
| | Engineering | $K_6 = 0.393598$ |
| | Supply of electricity | $K_7 = 0.186121$ |
| 2 | Agriculture | $K_8 = 0.259002$ |
| 3 | Construction | $K_9 = 0.313017$ |
| 4 | Trade | $K_{10} = 0.34641$ |
| 5 | Entrepreneurship | $K_{11} = 0.472871$ |
| 6 | Transport and communications | $K_{12} = 0.447214$ |
| | Education | K13 = 0.482057 |
| | Healthcare | K14 = 0.34641 |
| 7 | Culture | K15 = 0.156508 |
| | Sport | K16 = 0.141421 |
| 8 | Tourist and resort | $K_{17} = 0.186121$ |
| 9 | Foreign | $K_{18} = 0.313017$ |

Sources: authors' development.

In step 4, the developed methodology was applied to the DSTU specialties based on the data of the questionnaire for students and entrants, the data of the employment center of the city of Kamianske, and the data of the distribution of students of the first year of bachelor by specialties in the direction for 2010–2016. As a result of numerical processing, an optimal distribution of budget places (in percentage) for specialties DSTU is presented in Table 5.

**Table 5.** Optimal allocation of budget places.

| № | Specialties | Value, % | № | Specialties | Value, % |
|---|---|---|---|---|---|
| 1. | Management | 16 | 10. | Automobile transport | 4 |
| 2. | Software Engineering | 15 | 11. | Sociology | 4 |
| 3. | Applied Mathematics | 8 | 12. | Ecology | 3 |
| 4. | Engineering | 7 | 13. | Biotechnology | 3 |
| 5. | Philology | 7 | 14. | Chemical technology | 3 |
| 6. | Finance and Credit | 7 | 15. | Physics | 2 |
| 7. | Accounting and Auditing | 5 | 16. | Welding | 2 |
| 8. | Engineering mechanics | 5 | 17. | Metallurgy | 2 |
| 9. | Heat power engineering | 5 | 18. | Foundry production | 2 |

Sources: authors' development.

A balanced optimal distribution of budget places in the specialties of DSTU was obtained thanks to the practical application of the theory of system constraints and apparatus of fuzzy set theory.

### 5. Conclusions

The main results were as follows:

1.  The conducted analysis of primary and secondary data showed some differences in the assessment of the attractiveness of the specialties of the University for consumers of educational services and the demand for the relevant specialty in the labor market, in terms of the economic situation in the region and the proposal of universities and MES. Economic and humanitarian specialties are the most attractive for consumers, and the labor market requires more graduates of technical specialties. In this situation, the university, as a provider of educational services, must balance the demand of the consumer (entrants, students, and their parents) with the offers of the customer (state, entrepreneurs, labor market);
2.  Considering that a significant number of specialists who receive higher education are not employed on a specialty, and the demand of specialists in the labor market has not become a determining criterion for assessing the performance of the institution of higher education, it is necessary to allocate budget by taking into account the demand of certain professions among entrants and employers of the region and in the labor market;
3.  The method of the optimal distribution of the University budget by means of a practical application of the theory of system constraints and the theory of fuzzy sets was proposed. As a result, the distribution of budget by specialties was obtained in percentages. This will achieve an optimal balance between the demand for specialists with specific specialties and their proposal;

The practical significance of the obtained results is the implementation of the developed methodological provisions for forecasting the demand and supply of educational services of higher educational institutions. The complexity and interdependence of managerial problems in the higher education system require new ideas and approaches, which necessitates the search for new managerial decisions, namely the following:

1.  Forecasting the number of students, opening new specialties that are in demand in the labor market (in the DSTU there are opportunities for training teachers, teachers of mathematics, computer science, laboratory engineers for chemical and physical research);
2.  Implementation of the optimal allocation of budget places according to the proposed method, including the means of the theory of constraints of the system and the theory of fuzzy sets;
3.  The involvement of graduates of higher educational establishments and employers in an active introductory campaign and dissemination of information on the specialty of an educational institution and a specialty in demand in the labor market of the region, and proposals for future employment;
4.  The involvement of employers, representatives of state, and private enterprises of the region in the educational process—holding practical classes, organizing educational and industrial practices at the enterprises of the region with the aim of acquiring students certain experience and professional skills, developing tasks for coursework and diploma papers, and further supervising their implementation;
5.  The cooperation of higher educational establishments and enterprises for organizing and conducting competitions of students' scientific and practical works on problematic issues of these enterprises with the further employment of the winners of the competition;
6.  Conducting high school specialist tests to determine competency qualities and abilities among entrants in the region, active vocational guidance work aimed at informing about the prospects for the development of this region, and forecasts of the demand of specialists.

This study had several limitations. The survey was conducted among students from all departments of the DSTU studying different specialties, and graduates of Kamianske schools. The analysis of the research was conducted according to the demand for specialists in the labor



market of the Dnipropetrovsk region and the data of the city employment center. The developed methodology was applied to specialties of Dniprovsky State Technical University of Kamianske on the analyzed results of research on the specified region. The proposed methodology can be applied to determine the optimal allocation of budget places of higher education institutions and forecast the demand and supply of educational services after conducting a similar study in a particular region and educational institution.

The tasks for further research on this issue are the implementation of the forecasting of the demand for specialists for the future in the region and the country as a whole, taking into account the development of innovative technologies, investment programs through systems analysis, and simulation modeling. The forecasted results of demand can be used as one more factor, in addition to the demand for specialists in the labor market at the present time and demand among entrants and students to determine the optimal allocation of budget places in the specialties of the educational institution. Completed forecasting of demand for specialists will also allow the study of prospects for opening new specialties in higher educational institutions and qualitative training of faculty, material, and technical base for the training of qualified specialists.

**Author Contributions:** Formal analysis, N.D.; Methodology, I.K.; Software, Supervision N.V.; Writing—original draft, N.S.; Writing—review & editing, I.G.

**Funding:** This research received no external funding.

**Conflicts of Interest:** The authors declare no conflict of interest.

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
