# Peer review of "Strategic Imperatives of Managing the Sustainable Innovative Development of the Market of Educational Services in the Higher Education System"

_sustainability, doi:10.3390/su11247253_

Round 1
Reviewer 1 Report
Strategic Imperatives of Managing the Sustainable Innovative Development of the Market of Educational Services in the Higher Education System
The paper presents empirical research on the main trends of the development of the market of educational services and the labour market in Ukraine.
The English proof-reading MUST BE DONE! The abstract should be improved and have to provide more structured aim, scope and background, to state the principal objectives and scope of the investigation, to describe the methods employed, to summarise the results, to state the principal conclusions, recommendation and outlook. Please adjust the abstract according to the following structure: purpose, methodology, findings, research implications (if applicable), practical implications, the originality and value of the paper. The volume of an abstract should not exceed 200 words. In the introduction, the context of the research should be established, the purpose and/or hypothesis that was investigated should be stated. The information on previous research on the subject can be included either in Introduction or in the following section. Also, the main idea, importance, novelty, etc. can be indicated in this section The aim of the paper should be clearly stated in one sentence and presented in the introduction. Please briefly describe in the last paragraph of the INTRODUCTION section, the content of each section of the paper and include brief information on methods (one sentence). The main text should include previous research on the subject, methodology or theoretical framework, results of the research, and discussion with the interpretation of results obtained. Please provide the theoretical grounding of the research. The Literature review should trace the intellectual progression of the field, give an interpretation of previous research, identify where gaps exist in how a problem has been researched to date. From 17 references in the reference list, only 3 of them reflect research made in the recent five years and only a few of them published in the high-ranking journals abstracted in the Clarivate Analytics Web of Science or Scopus databases. Please provide the literature analysis based on 25-30 recently published papers from high-level scientific journals indexed in Clarivate Analytics Web of Science or Scopus databases (years 2017-2019). METHODOLOGY. Please provide the answers to the questions: How was the data collected or generated? And how was it analyzed? The methodology section of your paper should clearly articulate the reasons why you chose a particular procedure or technique. The methodology should discuss the problems that were anticipated and the steps you took to prevent them from occurring. Provide background and a rationale for methodology. Discussion section must be separated and must compare obtained results with other authors. Please provide research LIMITATIONS. English proof-reading is highly recommended. Please check line 17 “…and practical recommendations for the development of a strategy for the development of a higher…”.Author Response
Thank you for your review.
1.The abstract is improved ans structured according to the recommendations.
2.The introductions is redone according to the structure specified.
3.The Material and Method section has been revised based on the above remarks and questions.
4.Literature review and References expanded.

Reviewer 2 Report
the article is interesting and brings lights to the field that belongsAuthor Response
Thank you for your review.

Reviewer 3 Report
The topic of the education system is important for the development of each organization in all economic environments. Authors focused on the specific situation in Ukraine higher education system, but the processed text has several errors:
The structure of the paper does not match the requirements for scientific papers. Literature research, which has to prepare a base for the whole theory background, is included in the RESULT chapter. It is necessary to extend the METHODOLOGY chapter, which has a lack of individual methodological description. Quotations in the paper reflect the Harvard style, which is not matching the style of the SUS journal, which requires figures in brackets i.g. [1]. The text is also a URL link (L148). The paper needs to get English proof of all text. In the text, there are several parts with no quotations, e.g. L142-151. It is not possible to present ratio by non-ration figures as authors present in FIGURE 3 (L.154). Titles of all tables are not according to normal requirements. References in the list are not according to the defined format.
Author Response
Thank you for your review.
1.The materials of sections Material and Method and Results and discussionare redistributed according to the above remarks.
2.The names of the tables and figures have been corrected.
3.Improved translation into English by a specialist translator.

Round 2
Reviewer 1 Report
The manuscript has been amended, but still, several points are left for improvement:
The main text should include previous research on the subject, methodology or theoretical framework, results of the research, and discussion with the interpretation of results obtained. Please provide the theoretical grounding of the research. Please add the separate section – LITERATURE REVIEW. It should trace the intellectual progression of the field, give an interpretation of previous research, identify where gaps exist in how a problem has been researched to date. From 22 references in the reference list, only 8 of them reflect research made in the recent five years, and only a few of them published in the high-ranking journals abstracted in the Clarivate Analytics Web of Science or Scopus databases. Please provide the literature analysis based on 25-30 recently published papers from high-level scientific journals indexed in Clarivate Analytics Web of Science or Scopus databases (years 2017-2019). Please provide research LIMITATIONS.Author Response
We corrected our article on your requirements.
